# Amalia Revisited—A Reanalysis of Amalia’s Dreams with the Method Structural Dream Analysis

**DOI:** 10.3390/brainsci13050796

**Published:** 2023-05-12

**Authors:** Christian Roesler, Daniel Widmer

**Affiliations:** 1Clinical Psychology, Catholic University of Applied Sciences Freiburg, 79104 Freiburg, Germany; 2Clinical Psychology, University of Zurich, 8050 Zurich, Switzerland

**Keywords:** dreams in psychotherapy, psychoanalytic dream theories, empirical and clinical dream research, Structural Dream Analysis, psychotherapy process research

## Abstract

Since Freud’s “*The interpretation of Dreams*”, working with dreams has continued to play a major role in psychoanalysis, though different perspectives have developed about the function and meaning of dreams. This controversy is discussed on the background of findings in empirical as well as clinical dream research. In this paper, the research method Structural Dream Analysis is introduced which investigates the changes in structure of dreams over the course of psychotherapy. The method is applied to the specimen case Amalia X, which is considered to be the best investigated case in the history of psychotherapy research. Based on the results from this case and from other studies, the implications for psychoanalytic dream theories, namely those of Jung and Freud, are discussed.

## 1. Introduction

Ever since Freud’s seminal work *The Interpretation of Dreams* [1], working with dreams has had a central status in psychoanalysis. The theories of the dream and its significance, though, as well as the use of dreams in therapy, have been controversial from the beginning, and have changed over time, as well as within various psychoanalytic schools. Furthermore, since the discovery of REM sleep, a substantial body of knowledge has arisen, as it has in clinical dream research; this knowledge may contribute to a better evaluation of various theories of the dream in psychoanalysis (see Roesler in print [2] for details).

Since the first formulation of a theory of dream interpretation by Freud, a number of controversies around understanding the meaning of a dream in psychoanalysis have developed and competing theories have been established. Whereas in Freud’s [1] perspective, the dream has the function of protecting the sleep and for that reason distorts the unconscious meaning of the dream, other theorists have questioned this idea of a censorship applied to the so-called latent meaning of the dream. Probably the most outstanding of the competing theories is that of Jung [3], who argues that, generally speaking, the dream provides a picture of the current situation of the entire psyche including unconscious parts and puts that into a spontaneously produced picture. In a second theory, Jung argues that the dream has the function to compensate the conscious attitude of the personality. Another idea, which Jung added to the theory of dreaming, is called the subjective level of interpretation: Jung assumes that in the dream, underdeveloped or even conflictual parts of the psyche (which is called a complex in Jungian theory) can appear in a personified mode; thus, in this perspective the most interesting question is about the relationship of the dream ego, which presents the conscious part of the personality, to the other elements in the dream which may be understood as representing complexes and other not yet integrated capacities of the psyche. Dream interpretation then focuses on the relationship of the ego to the other parts and elements in the dream, e.g., is the relationship one of cooperation or more of a threat.

In this paper, the research method Structural Dream Analysis is introduced which is based on a Jungian understanding of dreams and their meaning for psychotherapy processes. The method is applied to the specimen case Amalia X, which is considered to be the best investigated case in the history of psychotherapy research [4]. Based on the results from this case and from other studies, the implications for psychoanalytic dream theories, namely those of Jung and Freud, are discussed.

## 2. Empirical Dream Research

Apart from the psychoanalytic approaches to dreaming, there has been an independent tradition of empirical dream research since the discovery of rapid eye movement (REM) [5], which yielded notable empirical findings on the functions of dreaming and its connection to waking life.

According to Hall and Van de Castle [6], who created the most widely used system of content analysis for dreams, it is possible to draw a personality profile based solely on the dreams of a person. In the same methodological tradition, Levin [7] notes that the themes in dreams of a person show substantial continuity over time, while Cartwright [8] was able to show that the themes in the dreams change when a patient undergoes psychotherapy. Barret [9] researched dreams in patients with dissociative identity disorder and demonstrated that the split-off parts of their personalities appear personified in their dreams.

Investigating dreams of patients in psychoanalysis, Greenberg and Pearlman [10] found that the topics of the dreams was strongly connected to the themes currently discussed in the therapy sessions, so that the dreams represented the complexes/psychological conflicts in the current life of the dreamer. These findings were confirmed by research conducted by Palombo [11], who could demonstrate that the contents from the last therapy session were taken up in the following dreams of the patients. Popp, Luborsky and Crits-Christoph [12], making use of the methodology of the Core Conflictual Relationship Theme (CCRT), demonstrated that the narratives that were produced in therapy correlated highly with the narrative patterns in the dreams representing the unconscious core conflictual themes. These findings support the continuity hypothesis, one of the most prominent theories in contemporary empirical dream research, which states that the modes and contents of mental functioning continue from waking life into sleep and are thus not fundamentally different (for a more detailed discussion see Roesler [13]).

In an overview of the findings of empirical dream research, Barrett and McNamara [14] point out that in the dream, the brain is in a mode in which it does not have to process new input but can use larger capacities for working on problems and finding creative solutions. Thus, the function of the dream is to take up experiences in waking life that are emotionally charged for the dreamer and reprocess them in a specific mode of brain functioning which is able to connect larger areas and functions of the brain compared to waking life consciousness and can thus produce new solutions to the psychological problems.

## 3. Jungian Theory of Dreaming and Dream Interpretation

One of the prominent theories of dreaming is the one presented by C.G. Jung [3]. Jung assumes that the psyche is capable of self-regulation, and that dreaming plays a role in this by spontaneously producing pictures which can be seen as providing an account of the psyche’s current situation in the form of symbols, including unconscious content. Referring to the findings of empirical dream research presented above, it could be argued that these support, by and large, the perspective taken by Jung, interestingly without having had the intention to do so.

Regarding dream interpretation, Jung describes two different approaches. In the first perspective, called the interpretation on the subject of level, the characters and elements of the dream are interpreted as representing parts or qualities of the dreamer’s personality. This perspective is the focus of the present paper, as the idea behind it assumes that especially unintegrated parts of the personality, complexes and unconscious conflicts, appeared personified or in the form of pictures in the dream. The second perspective, called the interpretation on the objective level, assumes that the characters presented in the dream present persons or events in the waking life of the dreamer.

The general idea in Jung, and the reason why he uses working with dreams in psychotherapy, is that the unconscious tries to communicate with consciousness by the way of dreams. In the dreams, the unconscious presents parts of the psyche to the conscious mind which are not yet integrated into the whole of personality, points to problems or even presents solutions. Through dreams, it brings new information to consciousness, which can then be integrated if a conscious understanding of the information is possible.

Therefore, Jungian dream analysis lays its focus on the relationship of the dream ego (i.e., the figure in the dream which experiences the dreamer as “myself”, psychoanalytically representing ego consciousness) to the other actors in the dream. Through this imagery one gets an indication of the capability of the ego to cope with emotions, impulses and complexes (being represented in this symbolic form in the dream), and the strength of ego consciousness. Since the language of dreams is by means of symbols and images, a translation is needed to be understood by the conscious ego.

Contemporary dream conceptualizations based on empirical research strongly challenge Freud’s classical dream theory and dream interpretation assumptions: Firstly, there is no evidence of a distortion process leading to a difference between manifest and latent meaning, nor is the dream “the keeper of the sleep” etc. [15]. In recent decades, a reconceptualization of psychoanalytic dream theory has developed, influenced by findings from empirical dream research. The presented evidence has led to a convergence of contemporary Freudian dream theories moving towards a Jungian understanding of the dream (e.g., [7,16]). In reaction to these empirical findings, some modern Freudian dream theories have incorporated a number of aspects of Jung’s dream theory. J. Fosshage [16,17], is an example whose dream theory focuses on the functions of the dream as a regulator of emotions and integrator of psychological organization. Scholars and researchers in the Freudian tradition, however, still argue (in the face of empirical evidence that contradicts these Freudian hypotheses) for a process of dream censorship—which results in a distortion of the latent content—and also for the theory of wish fulfilment of dreams. According to these researchers, the overall function of dreaming is still supposed to be protecting the sleeper from being woken up by repressed impulses [18,19,20].

## 4. Overview of the Research Project “Structural Dream Analysis”

In the tradition of clinical dream research in psychoanalysis [21], coding systems have been developed (e.g., Hall and Van de Castle [6], Moser and von Zeppelin [22]) which aim at coding the content of dreams and thus make research on the process of therapy possible. This research found evidence for typical patterns of change in dreams over the course of therapy. For example, it was found that the dominating affect in the dream changes parallel to the therapeutic process and the development of the personality [23,24]. A problem in this tradition of research is that the methods applied are often based on theoretical elements taken from psychoanalysis which are taken for granted and can therefore not be tested in this kind of research.

Therefore, in the research project Structural Dream Analysis (hereafter, SDA), we attempted to keep the methodology free of any theoretical presuppositions, and instead chose a structuralist perspective to the investigation of dreams [25]. This structuralist viewpoint starts from the assumption that the meaning of the dream is not transmitted in the first place by containing certain pictures or symbols (content), but in the relationship between the dream ego (the protagonist of the dream) and the other elements in the dream. In this methodological perspective, the focus is on the structure of this relationship, with a special focus on the agency of the dream ego, its capacity to deal with the other figures and elements in the dream and with the problem that is presented in the dream. The methodology provides a formalized approach to the interpretation of dreams independent of the interpreter; a number of tests found reliability coefficients of *k* = 0.70–0.82 four interpreter agreement on the same case/dream series.

SDA takes a narratological viewpoint on the dream, which means that a dream is seen as a narrative which presents a problem and attempts to solve the problem [26]. This makes the application of methods from narrative research possible (namely, Vladimir Propp’s [27] structuralist method of Functional Analysis and Boothe’s [28] narratological method JAKOB). It is assumed in psychoanalysis that dreams presented by the analysand in the course of therapy deal with the psychological problems of the dreamer and the attempts of the psyche to cope with these problems. Thus, changes in this structure of the dreams should be parallel to the improvement over the course of therapy. The aim of SDA is to identify this development in the structure of the dreams and compare it to the course and results of therapy. In the beginning, SDA was a complex research methodology, formalized in a manual, containing a systematic series of interpretive steps (for more details see Roesler [25]). This form of systematic analysis was applied to dream series covering the whole course of analytical psychotherapies.

The dreams are gathered by practicing analysts who also write a case report about the psychopathology and psychodynamics of the patient involved as well as about the development of core conflicts and themes in the course of the therapy. Clients are asked to keep a dream diary, but even if they do not, the analyst documents the dreams reported over the course of psychotherapy. Only when the dream series has been fully analyzed by using SDA are the results are compared to the therapist’s report. SDA allows systematic and objective analysis of the meaning of dreams produced by patients in psychotherapy. In particular, this method focuses on the relationship between the dream-ego and other dream characters and the activity level of the dream-ego. The following questions are considered: Is there a generalized structure in dream development in successful psychotherapy compared to unsuccessful psychotherapy? Is there a connection between a type of psychopathology, e.g., anxiety, and the symbols and structures in the dreams? Initially, 15 well documented cases were investigated by applying all the steps described above. The patterns characterizing the individual dream series were reconstructed from the material alone, in a bottom-up analysis (for a detailed account of this analysis see Roesler [24]). The analyzed cases were then compared in a cross-case-analysis and the typical patterns described below were extracted (for more details about this sample and a detailed account of how the patterns were reconstructed see Roesler [29]). This typology was then tested with a sample of 150 case reports (which included a documentation of the dreams reported in the course of these therapies) from the archives of the Jung Institute Stuttgart/Germany and confirmed. All of these cases were long-term psychoanalytic treatments ranging from 150 to 300 sessions with a frequency of 1 or 2 sessions per week. The number of dreams per case differed widely from 10 to more than 90. In all the therapies, dream analysis played a major role but depended on the number of dreams the patient could remember and present in therapy.

## 5. Findings of the First Phase of the SDA Project

Based on the analysis of the cases mentioned above, it was found that the majority of all the dreams could be described by a limited set of structural patterns. These patterns circle around the ego being confronted with a task, having to cope with a challenge/threat, or attempts to fulfill a plan. A scale of five sub-patterns was developed which describe a rising activity/agency of the dream ego (for details see Roesler [29]):

Pattern 1: No dream ego present.

Pattern 2: The dream ego is threatened (a differentiation is possible regarding the capacity of the dream ego to cope with the threat).

Pattern 3: The dream ego is confronted with a performance requirement (here, the initiative is not with the dream ego, but with another person/figure who confronts the ego with a task; a typical four is the examination dream).

Pattern 4: Mobility dream (The dream ego is traveling towards a specified or unclear destination, using different forms of transportation such as bicycle, car, bus, train, airplane, ship etc. This dream pattern can again be differentiated depending on how much agency the dream ego possesses, thus how much the dream ego is in control of the movement. The extent of agency of the dream ego can range from total disorientation and imprisonment to a continual movement towards the destination overcoming various obstacles).

Pattern 5: Social interaction dream (The goal of the dream ego is to make contact or to communicate with another person or figure in the dream. This can include sexual contact. This dream pattern can be differentiated depending on how active the dream ego is in its pursuit and how successful it is in achieving the desired contact. The dream ego might be ignored by others, criticized or ridiculed or it might be successful in making contact. In a special case of this pattern, the dream ego is aggressive and might even kill others. This reflects the wish of the dream ego for autonomy and separation). (In the latest version of this typology, this last aspect has been replaced, and we have added a sixth pattern called autonomy dream, in which the dream ego separates from others feeling confident and enjoys their independence—which can also have an aggressive character—marking the highest level of agency).

Following the establishment of the above typology, in the continuation of the research project, the dream patterns were used as a coding system. This implies a certain training of the raters which resulted in reliability coefficients ranging from 0.7 to 0.8. The rating asks for a strong reduction in the meaning of the dream and focusses on the agency of the dream ego, i.e., is the dream characterized by a threat to the ego, or, in contrast, by an ego taking the initiative in moving towards their own goals or creating satisfying relationships—in that sense, there are no overlaps; if the dream to be coded is very long and complex, it might be necessary to divide it into parts which are then coded separately. Usually, around 10% of the dreams in a series cannot be coded (NA). The focus on dream ego agency usually helps to clarify disagreement in different raters.

It was established before that the dream content and repetitive patterns in dream series are a reflection of the dreamer’s personality structure and psychological problems. These insights were confirmed by the general findings of the SDA project presented here. There was a strong correlation between the patient’s personality structure and the level of the dominating pattern at least in the first half of the psychotherapy process, i.e., the agency of the dream ego in the dominating pattern reflected the capacity of the patient’s psychological capacity to cope with complexes/inner conflicts. This capacity in psychoanalysis is called ego strength or maturity of the personality. This reflects the degree of integration of the personality, of ego consciousness and other parts of the psyche and the quality of ego functioning.

Common findings were that when psychotherapeutic interventions were successful, there were improvements in symptoms, psychological well-being, and emotional regulation (the following is a shortened summary of the extended version of the general findings presented in Roesler [28]). From a psychoanalytic point of view, this was reflected by a gain in psychological structure and ego strength which was paralleled by a transformation of the dream patterns. Whereas in the first half of the therapy process, the dominating pattern reflected the inability of the dream ego to cope with threats or challenges to fulfill a task, typically in a middle phase, the pattern changed to more successful actions of the dream ego. For example, in Pattern 2 dreams (in which the dream ego is threatened) the dream ego begins to confront the threat, to actively fight back or find ways to cope with and finally overcome the threat. This can also lead to the realization by the dream ego that earlier threatening figures are apparently not so dangerous and can become friends. Similar transformations can be found for Patterns 3 and 4. Whereas in typical Pattern 3, dreams of the first half of the process the dream ego fails to accomplish the requested task, this changes into successful attempts to deal with the task, or the pattern even disappears.

Pattern 4 (mobility): Usually, in the first half of the series, the dreamer does not get to his destination, takes the wrong bus or train, does not have a ticket, the road is closed, or the dreamer cannot drive the car. In some cases, the dream ego might even be trapped in a wall and unable to escape. Later in the series, this usually transforms into a dream ego that successfully achieves desired goals and controls its mode of transport. Pattern 5 (social interaction): Cases in which this pattern predominates are most often characterized earlier in the series by the dreaming ego’s failed attempts (or passivity) to engage in desired contact or communication with others—the dream ego gets ignored by others, others might forget about the ego’s birthday, or the dream ego is harshly criticized and devalued by others. Towards the end of the dream series, the dream ego becomes increasingly capable of creating satisfying interactions with others, including sexual encounters, and receiving care and support from others. If the therapeutic change is successful, Pattern 5 will increasingly dominate the second half of the dream series. That is, the dream ego becomes dedicated to creating the desired social interactions. This happens after the dream ego is able to overcome the recurring negative pattern of dreams of threats, movement failures, or negative exams.

These patterns of transformation were found only when therapists reported improvements on the symptom level and positive changes in personality structure. A continuation of repetitive patterns, on the other hand, was found in treatments that the therapists considered as failure or still in progress, without any change in the structure or type of pattern.

On the background of these findings, the following hypotheses were formulated (summarized in Figure 1, see below):dreams in which the dream ego is threatened can be understood as a reflection of a conscious personality struggling with unconscious conflicts, repressed needs and complexes;the threatening figures can be seen as symbolizing psychological problems, complexes or repressed impulses;if the dream series is dominated (at least in the first half of the series) by a threat-flight pattern, the personality of the dreamer is usually characterized by structural problems, an unstable ego with a low level of personality integration and unclear boundaries;patients whose dream series are characterized by mobility and social interaction dreams typically have more integrated personalities with a higher level of ego strength;in successful therapies, there is a typical pattern of transformation with dreams moving from patterns 1, 2 and 3 dominating the first half of the series, to patterns 4 and 5 characterized by a rising agency of the dream ego and more successful activities in following personal plans and creating satisfying social relationships;these typical changes in the structure of the dreams can be interpreted psychodynamically as a movement from an initially weak ego structure not capable of integrating threatening emotions, impulses and complexes towards a stronger ego structure, supported by the course of therapy, which becomes more capable to integrate initially unintegrated parts of the psyche; as a result, the ego increasingly becomes capable to exercise willpower, carry out plans, achieve goals, and express its needs in social interactions.

This also supports the hypothesis that dreams can be understood as images of the current state of the dreamer’s entire psyche, including unconscious aspects and processes inaccessible to the waking life consciousness. The psychological problem and the dreamer’s ego integrity state are symbolized both in the form of the image and in the form of the dream pattern. The explicit dream content clearly depicts the dreamer’s psychological situation, often even dramatically. Many dreams set the dreamer’s strongest fears in definite images, especially in the first half of the psychotherapeutic process. This supports earlier findings by dream researchers (e.g., [30]), pointing to the fact that meanings of the dream can already be found in the manifest content.

Broadly speaking, these findings support the continuity hypothesis of empirical dream research, in so far as the content of dreams is seemingly not distorted but can be seen as a picture of the current state of the relationship between ego consciousness and unconscious parts of the psyche [3,31,32,33]. On the other hand, there is no actual evidence of dream compensatory activity, as claimed by Jung. Jung’s first theory has more evidence that dreams provide a complete picture of the overall mental state, including aspects of the unconscious. In this sense, the dream function can be described as complementing rather than compensating for the picture by adding aspects of life that are inaccessible to the waking consciousness.

## 6. An Illustration of the Methodology with the Exemplary Case Amalia X

The methodology described above will now be illustrated with an application to a classical Freudian case, the so-called specimen case Amalia X, a Freudian psychoanalysis of over 500 sessions, fully documented on video and the subject of many studies [4,34,35]. As part of this analysis, the patient presented in total 96 dreams. These dreams were extracted and made accessible for research [34,36,37,38,39]. Kächele et al. [4], who documented the case and made it accessible for research, provide the following biographical and diagnostic information about the case:

The female patient (35 years of age at the beginning of therapy) suffers from a disorder of her self-esteem and recurring episodes of depression (ICD-10: F34.1 dysthymia). The restrictions on her self-esteem were connected with her hirsutismus—the virile growth of hair all over her body since puberty—which resulted in social insecurity. Her defensive mechanisms included a compulsion neurosis as well as symptoms of anxiety, erythrophobia (the fear of blushing). These problems had a negative effect on her ability to form personal relationships and even more on sexual relationships—at the beginning of therapy she had not had any heterosexual contacts. In her childhood, she experienced her father as being emotionally cold and obsessive, and mostly absent due to his professional responsibilities. She felt inferior to her two brothers, and it seems that there was a certain extent of parentification from the side of her mother to compensate for the absence of the father. Due to a hospitalization of her mother because of tuberculosis, the patient at the age of three had to live with her aunt and grandmother for several years. After completing high school, the patient attended university and strove to become a high school teacher, but broke off and joined a monastery, which she left again after a few years due to religious conflicts. She returned to university and finished her exams to become a middle school teacher.

A number of standardized measures were applied to document the success of therapy (Freiburger Persönlichkeitsinventar (FPI); Gießen-Test, self- and therapist-assessment). Results show a significant improvement, a reduction in psychosomatic symptoms, a stabilization of mood and self-esteem and arise in extraversion; these results remained stable in the follow-up [4].

Earlier studies of the case [34] found characteristic transformations over the course of the dream series: the later the session in treatment occurred, the more “(…) of the text of the dreams was attended to and worked over cognitively.” (p. 8). Merkle [40] found systematic changes in the later dreams compared to earlier ones regarding relationships, which became more friendly and tender, a development towards a more positive dream atmosphere and improvements in problem-solving.

## 7. A Re-Analysis of Amalia’s Dreams Using SDA

In the application of SDA to the dream series, the typology of dream patterns described above was used as a coding manual; the coding resulted in the following distribution of patterns:

Pattern 5: 26 dreams.

Pattern 4: 16 dreams.

Pattern 3: 18 dreams.

Pattern 2: 18 dreams.

Pattern 1: 3 dreams.

The typology was not applicable to 15 dreams (coded NA) which were excluded from further analyses. To compute a reliability coefficient (square-weighted Cohen’s kappa coefficient) the total of 96 dreams was coded independently by 2 raters, resulting in the following reliability: *κ* = 0.814, *p* < 0.001 (80% agreement). Since the case is considered exemplary and therefore successful (see also findings reported above), the hypotheses tested here predicted an increase in dream patterns from patterns 1, 2 and 3 towards patterns 4 and 5 over the course of Amalie’s therapy; in the same manner, a rise in agency of the dream ego was expected, meaning more activity and initiative of the dream ego, better coping with problems and more successful solutions/actions.

Figure 2 shows the dreams of Amalie X in sequence over the period of her therapy. Time equal 1 marks the first dream (6th session of therapy), the last dream number 96 (session 517). The descriptive scatter plot shows an overall association between the dream patterns and their occurrence in time, which was also found to be statistically significant (Kendall’s rank correlation, *r* = 0.25, *p* = 0.003). This is equivalent with a movement from pattern two to pattern five. If the dream series is divided into halves, with dream number 48 marking the middle point, it becomes obvious that the second half of therapy is characterized by a domination of pattern 5.

The division into two halves of the series was also used for testing the second hypothesis, stating that there is a rise in agency of the dream ego towards the end of therapy, which is equivalent with the dream ego overcoming threats and performing more successful actions. The association of succeeding/failing on the one hand and the first/second-half-of-therapy on the other hand was tested by Chi-square test and found to be significant (*X 2*(1, *N* = 79) = 5.2304, *p* = 0.022), thus supporting the hypothesis.

Since the dream series starts from patterns 2 and 3, and only 3 dreams were assigned to pattern 1, and is soon dominated by patterns 4 and 5, this also supports the above theory that in cases with a comparably stable personality structure and conflicts more on a neurotic level (as is the case with Amalia), the dream series is characterized by higher order dream patterns and only to a small extent by lower order patterns.

A qualitative analysis (making use of the method of amplification) of the symbols “mother” and “hair” found that the problems the patient had with finding an appropriate feminine and sexual identity and to stabilize her self-esteem were quite directly mirrored in the dreams and their symbols. In that respect, it may even be problematic to speak of “hair” appearing in the dreams as a symbol, since it is so obvious that it refers to her hirsutism. Beyond the midpoint of the dream series, there is a development clearly visible in which the patient becomes more able to deal with her sexuality and femininity in a more satisfied way. This goes parallel to the changes in her personality and relationships in the course of therapy. The therapist as well as the standardized measures clearly document that there is a gain in ego structure and self-esteem; over the course of the therapy, the patient became increasingly autonomous from her mother and was involved in more satisfying social relationships. The dreams mirror the process of therapeutic change which goes hand in hand with the integration of the formerly conflictual parts of the psyche.

## 8. Summary

The analysis of the 96 dreams of the case of Amalia with SDA clearly demonstrate that the transformations observed in the patterns and the agency of the dream ego in the dreams parallels the transformation of the patient’s personality structure and rising ego strength. This goes parallel with the findings of the earlier studies. For example, Albani et al. [41] investigating the dreams of the first half of therapy found that these are characterized by wishful thinking that others should be more attentive to the dream ego and that the self should be more self-determined, while on the other hand others were assumed to be unreliable and the response to the self is satisfying or even scaring.

Kächele et al. [4] summarize the findings of earlier studies on the case Amalia X focusing on changes in overall self-esteem: positive self-esteem increased significantly during the course of treatment, although the trend did not set in right at the start of treatment but only after wide fluctuations over the first 100 sessions; also, negative self-esteem showed a significant and continuous decrease from the beginning of treatment. The score of total suffering is characterized by a monotonic and statistically significant negative trend, meaning that “helplessness in dealing with suffering” decreased significantly over the course of treatment. Regarding dream emotions, there is a continuous pattern of change from negative dream emotions at the beginning to positive emotions towards the end of the analysis. The same positive development was found for problem-solving activity of the dreamer.

The findings of the present study also parallel findings of clinical dream research in general which found that dreams in successful psychoanalytic therapies are characterized by improvement/positive changes regarding dream atmosphere, an enlarged spectrum of affects present in the dream, successful problem-solving, and the movement of the dream ego from being less an observer to being more actively involved in the dream [42]. The latter was especially confirmed in studies on changes in the dreams of PTSD patients [43]. It was also found that there is a strong correlation between the agency of the dream ego in dreams on the one hand and the capacity of the dreamer in waking life to effectively cope with stress and regulate emotions [44].

## 9. Discussion

Thus, our study confirmed the assumption that dreams mirror the personality development and core conflicts appearing in psychotherapy. The developing ego strength of the client is reflected in the scope of action that the dream ego is able to initiate in relation to other figures and by the dream patterns described by SDA. Thus, the information about the personality structure is not just shown in static symbols and images but rather in patterns of the relationships between the dream ego and other characters in the dream. Moreover, the structural analysis of Amalie’s dreams showed absolutely no indication that the basic theme of all the dreams is a distorted, infantile wish. Quite the contrary, most of the dreams end in failure and address Amalie’s fear of rejection in a very dramatic and open fashion without any sign of distortion. The same applies to the sexual dreams. No distortion of the sexual content whatsoever can be found. We interpret these findings as providing support for Jung’s theory of dreaming and speaking against Freud’s basic assumptions: the dream seems to give a picture of the current situation of the psyche, including unconscious aspects, is without any distortion or covering up of a hidden content. Additionally, dream patterns change accordingly to the development the patient and his/her inner world takes over the course of therapy. However, it has to be noted that the material presented here does not allow for definite judgements about the dream theories competing in psychoanalytic thought.

## Figures and Tables

**Figure 1 brainsci-13-00796-f001:**
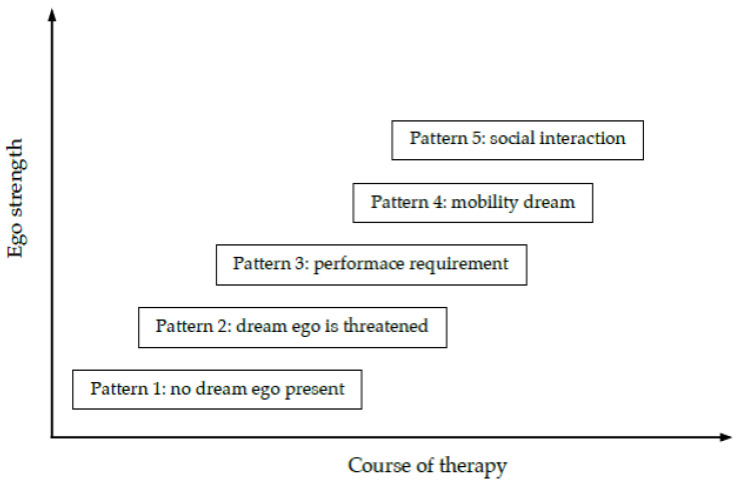
Changes in dream patterns over the course of therapy in connection with improvements in ego strength.

**Figure 2 brainsci-13-00796-f002:**
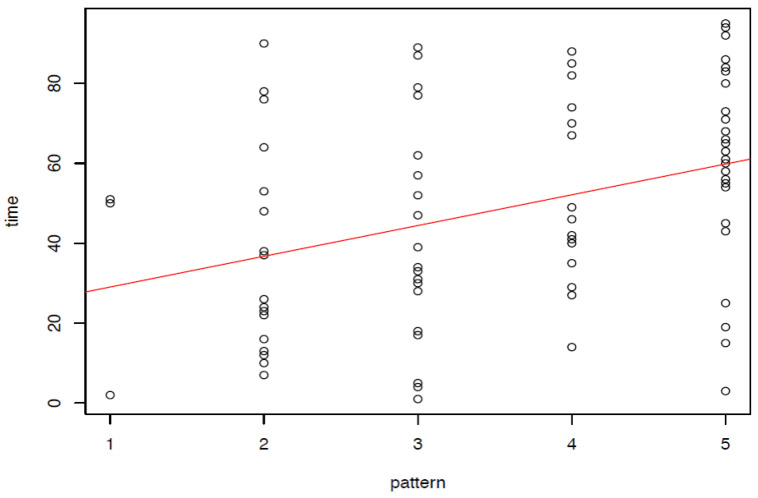
Amalia X, occurrence of dream patterns over time; scatter plot.

## Data Availability

Widmer, D. (2019). *Structural Dream Analysis: The Case of Amalie X* (Master Thesis, University of Zurich). Available online: www.infap3.eu, accessed on 9 May 2023.

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
