# Peer review of "Amalia Revisited—A Reanalysis of Amalia’s Dreams with the Method Structural Dream Analysis"

_brainsci, 2023, doi:10.3390/brainsci13050796_

Round 1

Reviewer 1 Report

Very well written, relevant and interesting study. The restricted space of the article does not allow an extensive theoretical, conceptual and methodological debate, so the authors present the method, the results of the study and the clinical implications  in a largely summarizing way. So, examples of dream reports and short examples of the interpretative procedure were not integrated. The theoretical Freud-Jung-opposition is inspiring, but I think that method and material create interesting questions but do not yet allow a validated  decision in favor to a Jungian Perspektive. 

Author Response

Some reflections were integrated which state that the material presented here does not allow for final judgments about the competing dream theories in methodologies of dream interpretation competing in psychoanalytic thought. A link to supplementary material was added which provides detailed accounts of the application of the methodology.

Reviewer 2 Report

The significance of this study is in the methodology used in analyzing dreams generated by patients. It provides a systematic and objective analysis of the dreams generated by patients in psychotherapies. The Structural Dream Analysis as an approach focus in particular on the degree of dream ego activity and its relationship to other dream characters.  As a scientific method that attempts to identify inner structure of meaning from a series of dreams the use of a systematic and transparent qualitative interpretive steps was deemed appropriate- in this way the researchers were able to make the application of narrative interpretation possible through the narratological method JAKOBS which was effectively described and applied in the manuscript. 

The manuscript could have benefited from providing a summary table that depicts the steps of interpretation -based on the symbols and the frequency of each of their occurrence. 

Although the SDA as a method is not new, its application in this study does extend the scientific validity as a methodological approach. 

Author Response

The table describing the steps of analysis is already provided in the text (172- 182). A more detailed description would expand the space of the paper. But a link to online material was added where a detailed account of the application of the method can be found.

Reviewer 3 Report

Thank you very much for the opportunity to review this paper. It provides both a theoretically and empirically significant contribution to psychoanalytic dream research. It summarizes the theoretical points of view in the required brevity in a very clear and understandable way and yields interesting empirical insights.

There are some minor issues and one major issue (point 4) that should be improved before the paper is published:

1. In section 2, you reference several empirical studies - it would be helpful for background understanding if you indicated on the basis of which data/methods of analysis the reported results were obtained.

2. Section 4: You raise two interesting research questions, but do not provide further details about the number of cases studied and the data available. I would be interested to know how the dreams were documented: were they summarized in retrospect by the analyst based on the session with the patient, transcribed audio recordings, etc.?

3. In section 5 you describe the different dream patterns: Are there overlaps? Can a dream belong to more than one pattern? I suspect that social interaction can also occur in mobility dreams or dreams with performance requirements. How did you manage to achieve a clear assignment?

4. Section 5 again raises questions about the methodological procedures on which these results are based: How many cases were analyzed? How many sessions, with which patients and in what setting (method, once/twice/three times a week) were conducted with the patients included in the study, and how many dreams per patient were analyzed to arrive at the results? What role did dream analysis play in the therapies? Were the results (changes in dream patterns) really as clear-cut as you describe? The results are very interesting, however there is little way to intersubjectively understand the analytic process that led to them. Also, you keep using frequency descriptions (most common...), but the reader has no idea on what they are based. Therefore, the strength of the hypotheses at the end of the chapter remains unclear.

5. Line 268: a sentence is repeated

6. In line 337 you state that “The findings show no evidence of a censorship process in the Freudian sense.” Please go into a bit more detail how this would have manifested in the data so that the reader is able to follow the argument. You argue further, that “the explicit dream content clearly depicts the dreamer's psychological situation, often even dramatically.” How did you arrive at this conclusion? Did you compare the content of the dream to the patient`s narrative from the sessions? I`d suggest being more explicit.

7. Your statement in line 341 that “findings also disprove the dream-fulfillment hypothesis” let me wonder how you selected the dreams analyzed in the study. Did patients report every dream they could remember or only those that contained a conflict or struggle? Might it be that fulfillment-dreams were not reported that often because they were not perceived as troublesome or problematic?

8. Section 8: How did you handle those cases, where coders disagreed?

Author Response

Referring to the points made by the reviewer, the following changes were introduced:

  1. some notes concerning the methodologies used in the studies were added.
  2. More details about the sample material and how the dreams were documented was provided in section 4.
  3. More information about the coding procedure was introduced into section 5.
  4. There is more detailed information provided regarding the samples etc. and the end of section 4. Nevertheless, the detailed description of the research process, how the methodologies applied and how the typology of dream patterns was constructed have already been published and cannot be repeated here but just summarized (see Roesler 2018b and 2018c for details). The aim of the present paper is to demonstrate the new step in the methodology, the testing of the hypothesis of a rising movement through the patterns parallel to improvement over the course of therapy. This is exemplified with a well-known and highly investigated case from the history of clinical research in psychoanalysis.
  5. The duplicate was deleted.
  6. + 7. The text discussing and criticizing Freudian concepts such as wish fulfillment and censorship were deleted. We agree with the reviewer that the discussion of the validity of those concepts would need a much more expanded discussion which goes beyond the scope of this paper respectively cannot be provided in the limited space given here.

  1. + 8. See information added in section 5 (264 to 273).